# Quality assured implementation of the Slovenian breast cancer screening programme

Katja Jarm[1,2], Maksimiljan Kadivec[1], Cveto Šval[1], Kristijana Hertl[1], Maja Primic Žakelj[1], Peter B. Dean[3,4], Lawrence von Karsa[4], Janez Žgajnar[1], Barbara Gazić[1], Veronika Kutnar[1], Urban Zdešar[5], Mateja Kurir Borovčić[1], Vesna Zadnik[1], Igor Josipović[1], Mateja Krajc[1,2]*

1 Institute of Oncology Ljubljana, Ljubljana, Slovenia, 2 University of Ljubljana, Ljubljana, Slovenia, 3 Department of Diagnostic Radiology, University of Turku, Turku, Finland, 4 Formerly at the International Agency for Research on Cancer, Lyon, France, 5 Institute of Occupational Safety, Ljubljana, Slovenia

* mkrajc@onko-i.si

**Data Availability Statement:** Table 1 contains the minimal data set upon which the present study is based. The data source (Breast Cancer Screening Registry DORA) is stated under the table.

## Abstract

### Setting

The organised, population-based breast cancer screening programme in Slovenia began providing biennial mammography screening for women aged 50–69 in 2008. The programme has taken a comprehensive approach to quality assurance as recommended by the European guidelines for quality assurance in breast cancer screening and diagnosis (4th edition), including centralized assessment, training and supervision, and proactive monitoring of performance indicators. This report describes the progress of implementation and rollout from 2003 through 2019.

### Methods

The screening protocol and key quality assurance procedures initiated during the planning from 2003 and rollout from 2008 of the screening programme, including training of the professional staff, are described. The organisational structure, gradual geographical rollout, and coverage by invitation and examination are presented.

### Results

The nationwide programme was up and running in all screening regions by the end of 2017, at which time the nationwide coverage by invitation and examination had reached 70% and 50%, respectively. Nationwide rollout of the population-based programme was complete by the end of 2019. By this time, coverage by invitation and examination had reached 98% and 76%, respectively. The participation rates consistently exceeded 70% from 2014 to 2019.

### Conclusions

The successful implementation of the screening programme can be attributed to an independent central management, external guidance, and strict adherence to quality assurance procedures, all of which contributed to increasing governmental and popular support. The

**Funding:** The author(s) received no specific funding for this work.

**Competing interests:** The authors have declared that no competing interests exist.

benefits of quality assurance have influenced all aspects of breast care and have provided a successful model for multidisciplinary management of other diseases.

## Introduction

Breast cancer is the leading cancer site in women in Slovenia and is a major public health problem. More than 1,300 women (125/100,000) are newly diagnosed with this disease yearly and the incidence is constantly increasing. Each year around 400 women (39/100,000) die from breast cancer [1]. Although opportunistic screening had been widely available to Slovenian women since 1990, only about 50% of all breast cancer cases were detected at a localized stage prior to the implementation of organised screening [2–4]. Opportunistic screening did not significantly improve earlier detection, and was associated with more operations for benign breast findings [5].

In a concerted effort to decrease breast cancer mortality rates and reduce the burden of this disease in the population, a program of systematic early detection through screening, effective diagnostic pathways, and optimised treatment with quality-assured services [6] was established.

Preparations for the organised screening programme began in 2003 and were coordinated by a team from the Institute of Oncology Ljubljana (IO) experienced in population-based cervical cancer screening [7]. It soon became clear that the healthcare system was not prepared for implementation of the screening programme using the existing analogue mammography units. There was a shortage of qualified radiologists and radiographers, and no standardised radiologic information system. To remedy the situation, the three institutional stakeholders in the programme, the IO, the Ministry of Health and the Health Insurance Institute of Slovenia (HIIS), all agreed to adhere closely to the standards and recommendations of the European guidelines for quality assurance in breast cancer screening and diagnosis (EU guidelines) [6]. Also, they strictly separated the organised screening programme from the symptomatic and opportunistic screening settings. Comprehensive protocols and procedures for the entire screening process extending from identification and invitation of eligible women to preoperative and postoperative assessment of women with screen-detected lesions were developed. The protocols and procedures were piloted in close consultation with international experts in implementation and quality assurance of population-based screening programmes. Readiness for expansion of screening volume and initiation of programme rollout was verified by a multidisciplinary team in 2008. This team, consisting of two radiologists, a medical physicist and the coordinator of the European Cancer Network based at the Screening Quality Control Group at the International Agency for Research on Cancer (IARC) in Lyon, France, inspected all components of the programme in operation. These included the coordination and evaluation unit, quality assurance of equipment, and all staff and facilities involved in screening, assessment, and management of screen-detected lesions.

The programme (named DORA) began inviting women from the Ljubljana municipality in March 2008 [8].

We describe here the organisation, the process of implementation and rollout, and the strengths and limitations of our screening programme. The aims of our study were to (i) assess the implementation process of the programme, (ii) assess quality assurance procedures and (iii) evaluate key performance indicators from the period 2008–2019.

## Materials and methods

### Screening programme

The breast cancer screening programme in Slovenia is organised, national and population based, with an explicit legislated national screening policy [4,9]. All screening methods and procedures are carried out in accordance with relevant guidelines and regulations. Informed consent is obtained from all subjects, including the use of their anonymized data for research. Only aggregate anonymized data was accessed for the preparation of the manuscript. All protocols were approved by the National Committee for Medical Ethics, the IO and the Slovenian Ministry of Health. Monitoring the programme with data collection and processing was authorised in the Healthcare database act [10]. The programme is funded by the HIIS, and all screening mammography, diagnostic tests and cancer treatment services are provided free of charge [11]. Coverage by Slovenian public health insurance applies to over 95% of the population and is required for participation in the DORA programme. The HIIS does not finance screening mammography performed outside the national programme (opportunistic screening) for women 50–69 years. The programme is centralized, with one semi-autonomous coordinating unit at the national level, situated at the IO where the multidisciplinary coordination team is also based (Fig 1). The advisory board of the programme prepares and amends the standard operating procedures (SOP) and guidelines with quality indicators, which all screening units (SU) are obliged to follow [12–14].

During the planning phase, all screening professionals received specialized training in reference screening centres abroad (Norway, Germany, Italy) or in training workshops conducted at the IO by international screening experts. After the first six years of experience, the IO became the reference training centre for all programme personnel.

### National breast cancer screening registry

The screening programme data is stored in a systematic population-based screening registry, the Slovenian breast cancer screening registry database (DORA IS), that collects and updates individual data, as recommended in the EU guidelines [6,15,16]. Comprehensive data on the target population is obtained from the Slovenian Central Population Register which is updated daily (complete information on age, vital status and residence). Additionally, the screening registry has a daily updated e-linkage with the Slovenian Cancer Registry database (information on breast cancer) and with the Register of Spatial Units of Slovenia (complete information on residence). Monitoring of the screening programme is facilitated by a dedicated software documentation system, specially designed for the DORA programme. Data at every step in the screening process is registered and accessible for evaluation through this software, which semi-automatically generates performance parameters and includes failsafe functionality in some modules.

The DORA IS is connected to the hospital and radiology information system at the IO and to the DORA Picture Archiving and Communication System (PACS). All screening images are stored in the central DORA PACS at the IO. All mobile and stationary SUs use dedicated software to send images to the DORA PACS through a wide area network (Fig 2).

### Invitation scheme

The programme invites women aged 50–69 at two-year intervals. There is no option for early (six-month) recall [6]. The exclusion criterion for invitation is a prior diagnosis of breast cancer (invasive or *in situ*). Personal invitation letters are sent from the IO, along with a specified date, time and place for the screening examination and an information leaflet about breast cancer screening. In the leaflet and in other communication to women the programme emphasizes the need to achieve an appropriate balance between the potential harms and benefits of

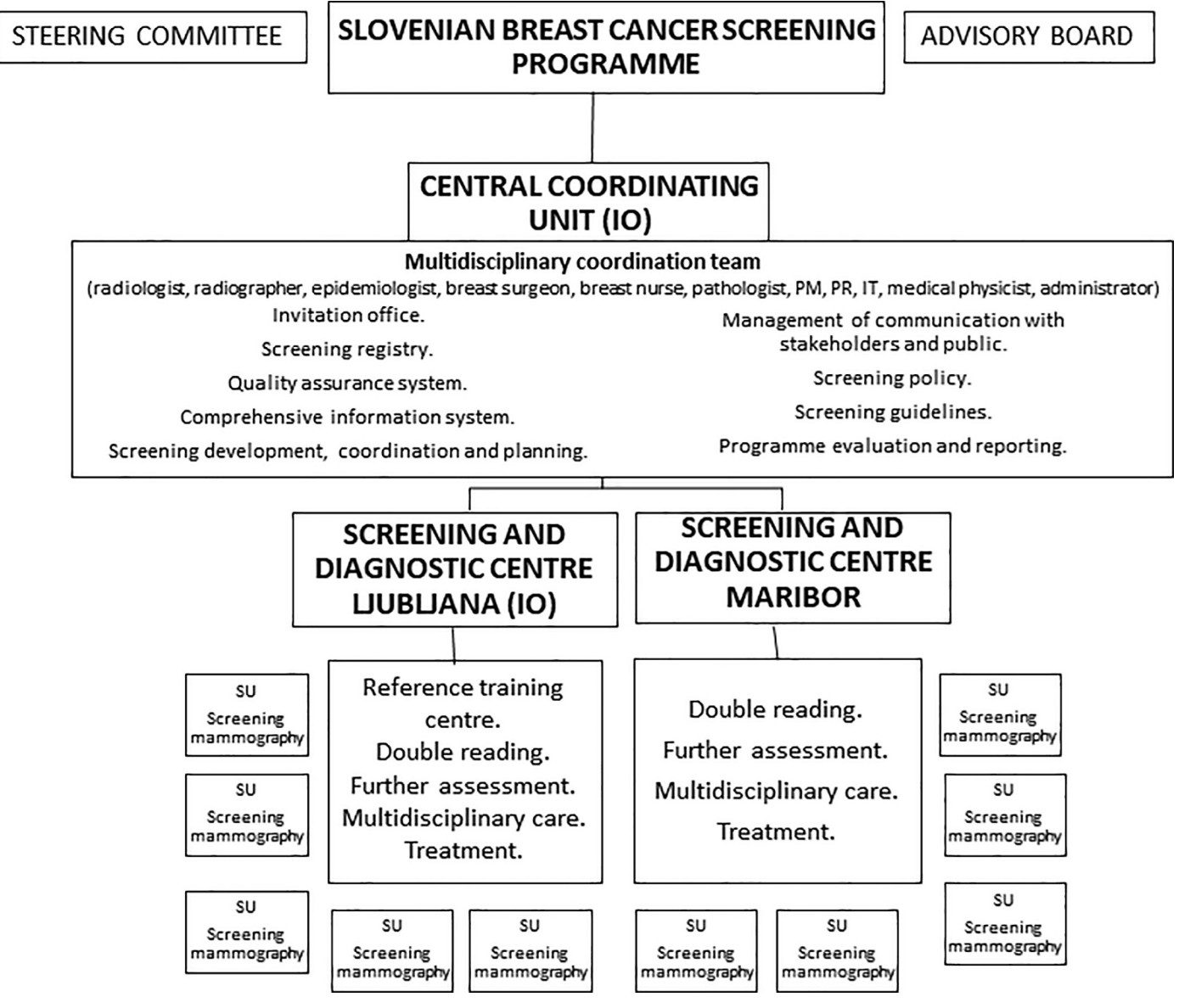

**Fig 1. Organisation of the Slovenian breast cancer screening programme.** IO–Institute of Oncology Ljubljana, PM–project manager, PR—public relations consultant, IT—information technology specialist, SU—screening unit.

breast cancer screening. The appointment can be changed on request via telephone or e-mail. Women are invited randomly to the initial round, and self-invitations from eligible women are also accepted. Women failing to respond to their invitation are sent one reminder after four weeks. If a woman still does not participate, she is invited again two years later. It is possible to refuse participation, which is also documented, after which the woman does not receive any further invitation from the programme, although she may rejoin at any time.

### Screening procedure

The stationary screening units are in the public healthcare setting (hospitals, community health centres), separate from cancer diagnosis and treatment. No other healthcare procedures are performed in the same office at the same time, so that healthy and asymptomatic women

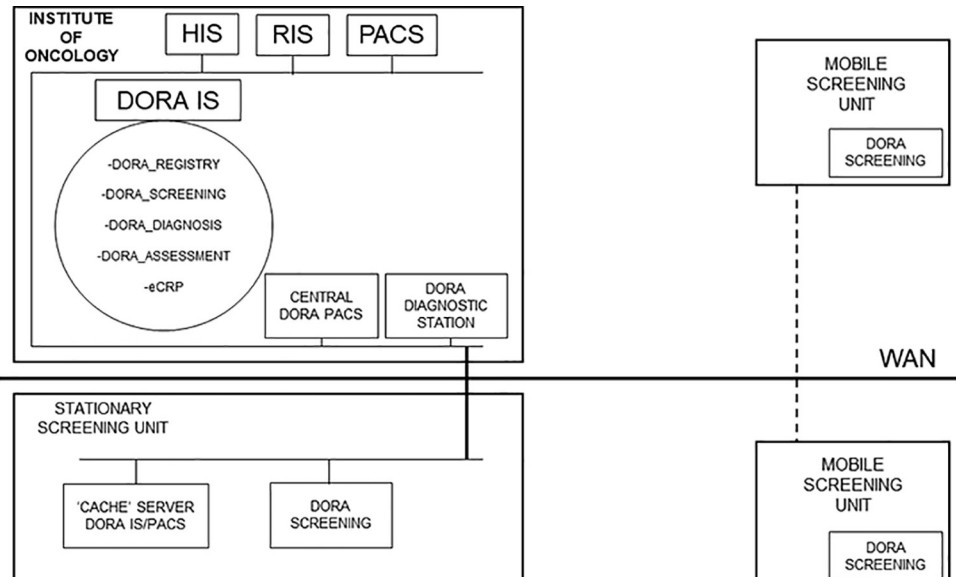

**Fig 2. Screening programme information system.** DORA IS—Slovenian breast cancer screening registry database, HIS–Hospital Information System, RIS–Radiology Information System, PACS—Picture Archiving and Communication System, WAN–Wide Area Network.

do not encounter breast cancer patients. Since 2018 a total of three mobile SUs have been screening women residing far from the stationary units. All SUs have been assigned to predetermined municipalities.

Upon arrival for her appointment at the SU, copies of the invitation letter and accompanying information materials are made available before a woman signs her informed consent statement prior to performing any procedures. The radiographer records a brief history of previous breast diseases/procedures and the results of an inspection for visible breast abnormalities (scars, skin changes). Clinical examination of the breast is not performed. The radiographer subsequently takes mammograms of both breasts in two views (craniocaudal, mediolateral oblique).

Independent double reading is done at the screening and diagnostic centres (SDCs) by two breast radiologists (readers), using the BI-RADS (Breast Imaging-Reporting and Data System) classification [17], excluding the ambiguous BI-RADS 3 category. The screening radiologists review all mammograms that have been interpreted by at least one of them as suspicious (BI-RADS 4a, 4b, 5) in consensus conferences with a third reader (lead radiologist). Letters informing women of their normal mammography results (BI-RADS 1 or 2) are mailed by post. The same letter also states that the woman will be re-invited after two years (Figs 3 and 4). The letter includes a recommendation to perform breast self examination (BSE) in the interval between two screening mammographies in order to raise breast awareness and avoid unnecessary delay in diagnosis and treatment of interval cancers. Women with abnormalities detected on the mammogram are informed of the result from the SDC by telephone and are given an appointment for further assessment. All screening test results are delivered within two to seven working days.

## Further assessment

Further assessment of screen-detected breast lesions is performed within four to ten working days after screening mammography at the SDCs, which are separate from other hospital

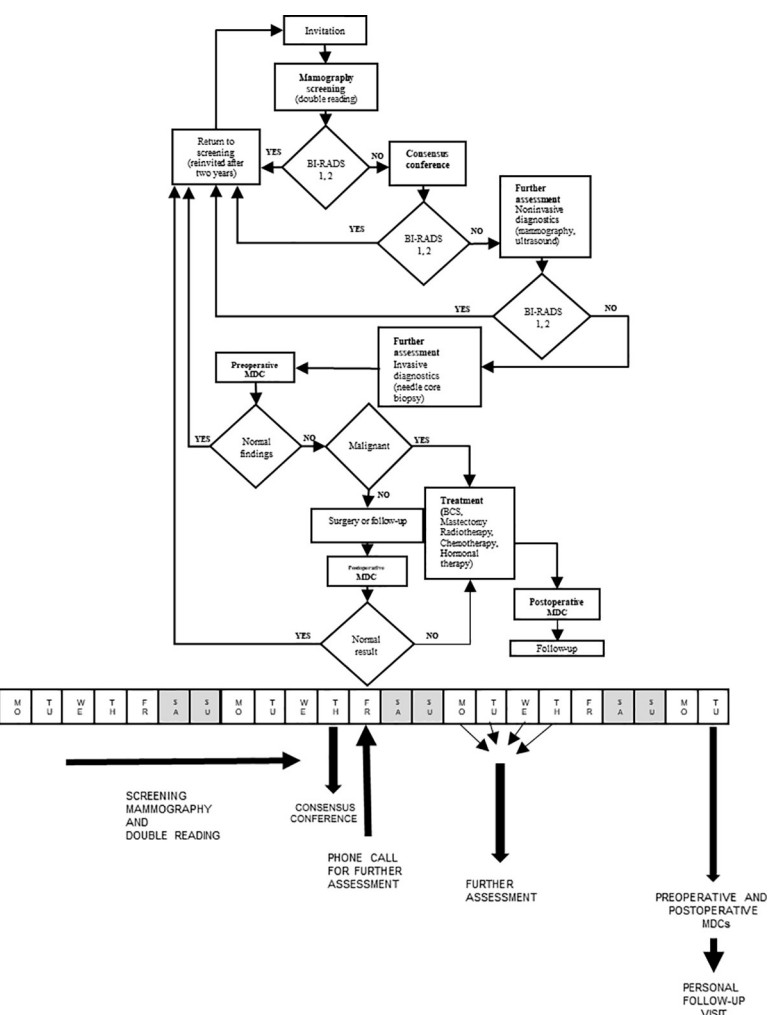

**Fig 3. Screening algorithm and timeline in the Slovenian breast cancer screening programme.** MDC–multidisciplinary conference, BCS–breast conserving surgery.

services. All appropriate assessment procedures, except MRI, are performed on the same day by a lead radiologist. Most of the suspicious findings are resolved by non-invasive procedures, including tomosynthesis, spot compression with microfocus magnification, and breast ultrasound. A benign result of these non-invasive examinations is communicated directly to the woman by a radiologist. If a breast biopsy is indicated, ultrasound-guided or stereotactic vacuum-assisted core needle biopsy is performed on the same day, and the result is communicated personally by a lead radiologist and a breast nurse at a follow-up visit within one week. The diagnosis of a breast lesion should be definitive; hence fine needle aspiration cytology and short-term follow-up are not practiced.

Diagnostic surgery is rarely performed; only if recommended by the pathologists and agreed upon at a multidisciplinary conference. Histopathology samples are double read by two pathologists. The SDC ensures that assessment and diagnosis are performed by a multidisciplinary team of dedicated radiologists, radiographers, pathologists, breast surgeons and breast nurses.

After core biopsy and after breast surgery, each woman's clinical, imaging and pathology findings are discussed and documented in weekly preoperative and postoperative

| POPULATION (SORS 2019H2) | Number |
|---|---|
| Slovenian population (female) | 2,053,381 (1,038,357) |
| Slovenian target population | 292,433 |
| Slovenian eligible population | 283,090 |
| **SCREENING MODE** | |
| screening target population | 50 - 69 years |
| screening test | digital mammography, two views |
| screening interval | 2 years |
| type of invitation | appointment (date, time, place) |
| **SCREENING CAPACITY** | Number |
| screening organisation | 1 |
| screening and diagnostic centre (double reading, consensus, assessment, treatment) | 2 |
| mammographic equipment | 21 |
| stationary screening units | 18 |
| mobile screening units | 3 |
| radiologists - readers | 18 (including lead radiologists) |
| lead radiologists | 9 |
| radiographers | 90 |
| other personnel at the screening and diagnostic centres including the managing unit at IO (nurse 8, administrator 15, IT 2, PHS 1, PR 1, MP 1, PM 1) | 29 |
| **PROGRAMME IMPLEMENTATION AND ACCEPTANCE** | |
| geographic extension by the screening programme | 100.0% |
| coverage by invitation | 97.7% |
| coverage by examination | 76.3% |
| participation rate | 78.1% |

| KEY POLICIES |
|---|
| Personal invitation for a specified date, time and place |
| CBE not used for Screening |
| Double reading of screening mammograms |
| Communication skills |
| High reading volumes |
| Mandatory minimum workload |
| Multidisciplinary meetings |
| Consensus conference |
| Further assessment management |
| Centralized further assessment |
| No cytology |
| Separation from symptomatic assessment |
| Double reading in pathology |
| Rapid delivery of results |
| Additional training of personnel |
| EU guidelines-based quality standards |
| Daily quality monitoring of mammographic equipment |
| Quality assurance for all personnel |
| Screening policy |
| Centralized programme management |
| SOPs and national screening guidelines |
| Screening registry |
| Central IT support and monitoring |
| e-linkage to registries (cancer, population) |
| Performance indicators reporting |
| Interval cancers monitoring |
| Volume-based reader compensation |

**Fig 4. Overview of the Slovenian breast cancer screening programme in 2019.** SORS 2019H2 –Statistical Office of the Republic of Slovenia, data on 31.7.2019, IO–Institute of Oncology Ljubljana, IT—information technology specialist, PHS—public health specialist, PR—public relations consultant, MP—medical physicist, PM—project manager, CBE–Clinical breast examination, EU–European, SOP–Standard operating procedures.

multidisciplinary conferences, respectively, with oncologists attending the postoperative conferences. If breast cancer is diagnosed, a woman is informed of this diagnosis within three weeks after screening mammography (Fig 3). Treatment of screen-detected breast cancers takes place at the medical centres where the SDCs are located.

## Quality assurance

Quality assurance has been prioritised since the beginning of the programme.

**Mammography equipment.** A dedicated centre for monitoring technical quality was established. The centre developed a special web application (ORQA: Online Radiological Quality Assurance) for daily data acquisition and monitoring of indicators recommended by the EU guidelines [6] for mammography devices and other technical equipment used in screening.

**Radiographers.** Radiographers undergo a multidisciplinary training programme for screening with a theoretical and practical positioning course and a refresher course every two years. Each radiographer performs a minimum of 36 screening mammography examinations per week. Image quality is audited by two lead radiographers at the reference centre with positioning evaluation twice a year [18,19].

**Radiologists, surgeons and pathologists.** Radiologists, surgeons and pathologists undergo additional training, and their performance is monitored using quality indicators for diagnostics and treatment. Quality assurance of mammography reading includes independent double reading, consensus conferences and routine audits of the specificity and sensitivity of each reader. An additional group review of interval cancers takes place every 6 months [19]. Each radiologist reads a minimum of 5,000 screening mammographic examinations per year. All pathology specimens are double read.

**Performance indicators.** Programme monitoring includes the following quality indicators selected from the EU guidelines, covering critical steps in the screening process from invitation to treatment:

- proportion of invited women who attend screening

- proportion of women recalled for further assessment

- proportion of women with invasive screened-detected cancer

- breast cancer detection rate

- proportion of screen-detected cancers that are stage II+

- proportion of invasive screened-detected cancers that are ≤ 10 mm in diameter

- time (in working days) between screening mammography and communicating results

- time (in working days) between decision to operate and date offered for surgery

## Data analysis

Evaluation of programme implementation in this communication is based upon the individual data from women invited between April 2008 and the end of 2019.

Population data was obtained from the Statistical Office of the Republic of Slovenia [20]. The target screening population was defined as women residing in Slovenia who were aged 50–69 in the index year. Because the target population was screened during a 2-year interval, the population data was divided by 2 to determine the annual population. Data from the Slovenian cancer registry was used to determine the eligible screening population after excluding women with a prior diagnosis of breast cancer [21].

Coverage by invitation was calculated as the number of invited women divided by the eligible population in the index year (Table 1). Coverage by examination was calculated as the number of screened women divided by the eligible population in the index year [6,22]. Extension of the screening programme was defined as the proportion of the national target population that resided in the areas in which the organised screening programme was up and running [23]. The ten geographic regions in Slovenia are divided into 212 municipalities. To calculate extension, the numerator was the target population of women in the municipalities in which screening was up and running in the index year, and the denominator was the national target population in the index year.

The participation rate was defined as the proportion of women invited for screening in a specific time period who were screened in the programme [6]. It was calculated from the number of invitations sent in the index year excluding reminders (denominator) and the number of women screened in response to those invitations (numerator).

## Results

### Programme implementation

The Slovenian nationwide programme was up and running in all screening regions within 15 years of the beginning of programme planning in 2003. The initial proposal for the screening organisation included six mobile SUs and two SDCs. Invitation to screening started in 2008, and before 2013 the programme was running only in the central Slovenian region. Financial constraints caused the three stakeholders to modify the programme, choosing digital

**Table 1. Rollout of the Slovenian breast cancer screening programme, 2008–2019.**

| Year | Number of mammography devices (end of the index year) | Number of regions performing screening | Extension of the screening programme [6]/[5] | Annual target population (women aged 50–69) | Annual target population (women aged 50–69) in regions screening | Annual eligible population (women aged 50–69) | Number of women invited | Number of invited women screened | Number of women invited to their first round of screening | Number of women invited to a subsequent round of screening | Proportion of invited women also invited to subsequent rounds [11]/[8] | Coverage by invitation [8]/[7] | Coverage by examination [9]/[7] | Participation rate [9]/[8] |
|---|---|---|---|---|---|---|---|---|---|---|---|---|---|---|
| [1] | [2] | [3] | [4] | [5] | [6] | [7] | [8] | [9] | [10] | [11] | [12] | [13] | [14] | [15] |
| 2008 | 1 | 1 | 14.4% | 130.359 | 18.813 | 127.252 | 2,793 | 1,844 | 2,793 | 0 | 0.0% | 2.2% | 1.4% | 66.0% |
| 2009 | 1 | 1 | 17.4% | 132.012 | 23.024 | 128.735 | 3,464 | 3,268 | 3,464 | 0 | 0.0% | 2.7% | 2.5% | 94.3% |
| 2010 | 3 | 1 | 19.7% | 133.528 | 26.258 | 130.099 | 12,804 | 9,868 | 11,982 | 822 | 6.4% | 9.8% | 7.6% | 77.1% |
| 2011 | 3 | 1 | 24.0% | 135.146 | 32.381 | 131.532 | 29,815 | 19,896 | 26,460 | 3,355 | 11.3% | 22.7% | 15.1% | 66.7% |
| 2012 | 3 | 1 | 27.3% | 136.302 | 37.259 | 132.531 | 40,210 | 25,348 | 32,260 | 7,950 | 19.8% | 30.3% | 19.1% | 63.0% |
| 2013 | 5 | 2 | 33.3% | 137.509 | 45.824 | 133.598 | 37,455 | 25,641 | 18,643 | 18,812 | 50.2% | 28.0% | 19.2% | 68.5% |
| 2014 | 8 | 2 | 41.4% | 138.809 | 57.433 | 134.782 | 42,991 | 32,641 | 21,789 | 21,202 | 49.3% | 31.9% | 24.2% | 75.9% |
| 2015 | 10 | 2 | 46.6% | 141.521 | 65.923 | 137.333 | 50,610 | 39,894 | 24,083 | 26,527 | 52.4% | 36.9% | 29.0% | 78.8% |
| 2016 | 14 | 6 | 72.5% | 144.258 | 104.547 | 139.872 | 75,436 | 57,121 | 42,369 | 33,067 | 43.8% | 53.9% | 40.8% | 75.7% |
| 2017 | 17 | 10 | 100.0% | 145.296 | 145.296 | 140.773 | 98,544 | 70,036 | 62,988 | 35,556 | 36.1% | 70.0% | 49.8% | 71.1% |
| 2018 | 20 | 10 | 100.0% | 146.052 | 146.052 | 141.380 | 120,087 | 90,390 | 70,988 | 49,099 | 40.9% | 84.9% | 63.9% | 75.3% |
| 2019 | 20 | 10 | 100.0% | 146.217 | 146.217 | 141.545 | 138,358 | 108,046 | 71,675 | 66,683 | 48.2% | 97.7% | 76.3% | 78.1% |

**Source:** Breast Cancer Screening Registry DORA, www.onko-i.si/eng/sectors/epidemiology-and-cancer-registry/registry-dora.

mammography units already available in public health institutions rather than purchasing additional mobile SUs.

The largest expansion of the programme occurred from 2015 through 2017 after this new strategic plan was adopted [13]. Extension of the programme reached 100% by December 2017, when screening units were up and running in all regions (fourteen stationary SUs—all part of the public health infrastructure—and two mobile SUs) and initial invitation had begun in the remaining municipalities. A third mobile and two stationary SUs were added in the beginning of 2018 due to a foreseeable need for additional capacity. By this time the screening infrastructure was already capable of inviting the entire eligible annual population (approximately 140,000 women) and coverage by invitation and examination had both increased to 85% and 64%, respectively. Programme rollout was complete in December 2019, when coverage by invitation and examination had reached 98% and 76%, respectively. The SDC at the IO has been in continuous operation since 2008, and a second SDC was established in Maribor in 2018 [24–26].

The programme development went through all stages of implementation: planning phase (2003–2008), pilot phase (2008), rollout ongoing (2008–2019), rollout complete (2019). It took 17 years to attain full invitational coverage. Gradual rollout and programme features are shown in Table 1 and Fig 4.

## Programme acceptance

The extent to which the programme covered the target population by invitations sent and by examinations conducted is shown in Table 1. Well before coverage by invitation reached 97.7% in 2019, the participation rate consistently exceeded the acceptable level recommended in the EU Guidelines (>70%). In 5 of the 6 years from 2014 to 2019 the participation rate exceeded the desirable level recommended in the EU Guidelines (>75%) [6].

## Discussion

The nationwide breast cancer screening programme was up and running across the country and all components of the programme were fully implemented at the end of 2017, within ten years after the first invitations had been delivered. By that time, all necessary screening institutions and units, technical equipment, trained personnel, the managing team, quality assurance protocols, legislation and screening registry were fully operational and able to offer organised screening to all eligible women. Given the complexity of the implementation process and the challenges of comprehensive quality assurance, 10 or more years are commonly required to establish organised population-based screening programmes [27]. This has been observed in various European countries such as the Netherlands (1988–1997) [28,29] and Denmark (1991–2007) [30] and now also in Slovenia.

The increases in programme coverage did not closely follow the increases in programme extension (Table 1). Regions were enrolled gradually by municipalities and the new screening units did not initially operate at full capacity. Furthermore, the proportion of women invited for subsequent rounds did not increase as rapidly as the programme extension. The current strategic plan was adopted in 2015, after which the rate of expansion increased owing to new directions and regulations and substantial support from the Ministry of Health and the HIIS. Eight additional regions were subsequently enrolled, increasing the proportion invited in the prevalent round.

It was important to sustain a high level of quality assurance during the implementation process even when it required slowing down programme rollout. Performance of all units was carefully monitored and audited to obtain the information required for the timely action to

maintain high quality. We also communicated this information to women attending screening in order to instil confidence in the programme and encourage participation in subsequent rounds.

Adequate and stable financing of the Slovenian central coordinating unit has been key to achieving and maintaining a high level of quality assurance in the programme. Preliminary results of surrogate indicators are promising [31] and suggest that the programme will have a discernible impact on breast cancer mortality.

## Strengths

DORA provides one of the best documented services in the Slovenian health care system, helping to improve the quality of breast cancer services throughout the country outside of the screening setting. DORA fulfils key requirements of the EU guidelines for well-organised cancer screening programmes as also emphasised in the second European screening report [6,32–35]. The programme benefits from:

i. screening policy and protocols specifying management procedures.

ii. a national screening organisation with one dedicated management team responsible for programme implementation, training, and auditing.

iii. earmarked public funding.

iv. standard operating procedures (SOP) with quality indicators established by national legislation and based on the EU guidelines, which have been followed from the beginning.

v. the legally mandated specification of the target population, the screening test and interval, and the protocol for inviting eligible women with appointments specifying time, date, and place.

vi. standards and supervision to ensure uniform screening conditions and performance in all SUs.

vii. the national population register which is updated daily and covers the entire target population.

viii. a comprehensive information technology system with e-linkage to the appropriate registries.

ix. systematic follow-up including monitoring of screening participation, performance indicators and long-term outcomes.

x. dedicated, comprehensive quality assurance protocols for radiographers, radiologists and for testing mammography equipment.

xi. streamlined, centralised further assessment, without the need to rely upon a GP for coordination.

## Weaknesses and barriers to screening

During the implementation period some barriers prolonged programme rollout and limited the potential effectiveness of the programme. There was a limited supply of trained personnel and available equipment, and the financial crisis of 2008 restricted available resources. The changes in the original screening organisational concept led to a more complex implementation, since more health institutions were included than originally planned.

Some women and their health care providers were reluctant to discontinue the practice of annual clinical breast examination which often included a referral to opportunistic screening. The same problem has been encountered in other European countries [36].

Prior to 2016, nonparticipants were not re-invited for subsequent rounds. The rationale for this decision was to gradually introduce screening by initially focusing on women who were favourably disposed to screening. This practice was discontinued.

## Conclusions

The Slovenian breast cancer screening programme has endeavoured to provide high quality diagnostic and therapeutic services, avoiding unnecessary procedures and delays, with equal access for all women in the eligible population. From the outset the programme has adhered closely to the key standards and recommendations promoted in the EU quality assurance guidelines [6] and in current European recommendations. After a 15-year implementation period beginning with comprehensive planning and piloting all components of the programme were fully developed, providing high quality screening and diagnostic services to eligible women throughout the country. Implementation could have taken longer in the absence of international collaboration and guidance during the planning and pilot phase. The complexity of screening services necessitated a gradual rollout of the programme. The benefits of quality assurance have influenced all aspects of breast care and have provided a successful model for multidisciplinary management of other diseases.

## Acknowledgments

The key scientific and professional support of the following experts and expert groups in planning and early implementation of the Slovenian breast cancer screening programme is gratefully acknowledged: Ms. Hildegard Aust, Wiesbaden, Germany (radiography); Dr. Margrit Reichel, Wiesbaden, Germany (radiology); Ms. Jutta Pfeiffer, Geneva, Switzerland (programme management and quality assurance); Dr. Per Skaane, Oslo, Norway (radiology); Dr. Martin Thijssen, Nijmegen, The Netherlands (physics); Dr. Blanka Mikl Mežnar, Ljubljana, Slovenia (former official at the Ministry of Health) and Dr. Mojca Senčar (deceased), Ljubljana, Slovenia (former president of Europa Donna Slovenia); Dept. of Cancer Screening and Unit of Cancer Epidemiology, Piedmont University Hospital, Turin, Italy; Quality Assurance Group, International Agency for Research on Cancer, Lyon, France.

## Author Contributions

**Conceptualization:** Katja Jarm, Maja Primic Žakelj, Peter B. Dean, Lawrence von Karsa, Mateja Krajc.

**Data curation:** Cveto Šval.

**Formal analysis:** Katja Jarm.

**Investigation:** Katja Jarm, Cveto Šval.

**Methodology:** Katja Jarm, Peter B. Dean, Lawrence von Karsa, Mateja Krajc.

**Project administration:** Katja Jarm, Maksimiljan Kadivec, Mateja Krajc.

**Software:** Cveto Šval.

**Supervision:** Maksimiljan Kadivec, Kristijana Hertl, Peter B. Dean, Lawrence von Karsa, Mateja Krajc.

**Validation:** Maksimiljan Kadivec, Kristijana Hertl, Peter B. Dean, Lawrence von Karsa, Janez Žgajnar, Barbara Gazić, Veronika Kutnar, Urban Zdešar, Mateja Kurir Borovčić, Vesna Zadnik, Mateja Krajc.

**Visualization:** Katja Jarm, Cveto Šval.

**Writing – original draft:** Katja Jarm, Mateja Krajc.

**Writing – review & editing:** Katja Jarm, Maksimiljan Kadivec, Kristijana Hertl, Maja Primic Žakelj, Peter B. Dean, Lawrence von Karsa, Veronika Kutnar, Urban Zdešar, Mateja Kurir Borovčić, Vesna Zadnik, Igor Josipović, Mateja Krajc.

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
