## [Decision Letter · Decision Letter 0]

5 Jun 2021

PONE-D-21-04831

Quality assured implementation of the Slovenian breast cancer screening programme

PLOS ONE

Dear Dr. Krajc,

Thank you for submitting your manuscript to PLOS ONE. After careful consideration, we feel that it has merit but does not fully meet PLOS ONE’s publication criteria as it currently stands. Therefore, we invite you to submit a revised version of the manuscript that addresses the points raised during the review process.

Please insert comments here and delete this placeholder text when finished. Be sure to:

Please revise your manuscript to address the reviewer's comments.

We look forward to receiving your revised manuscript.

Kind regards,

Rashidul Alam Mahumud, MPH, MSc, PhD

Academic Editor

PLOS ONE

Journal Requirements:

Reviewers' comments:

Reviewer's Responses to Questions

**Comments to the Author**

1. Is the manuscript technically sound, and do the data support the conclusions?

Reviewer #1: Yes

Reviewer #2: Yes

2. Has the statistical analysis been performed appropriately and rigorously? 

Reviewer #1: Yes

Reviewer #2: Yes

3. Have the authors made all data underlying the findings in their manuscript fully available?

Reviewer #1: Yes

Reviewer #2: Yes

4. Is the manuscript presented in an intelligible fashion and written in standard English?

Reviewer #1: Yes

Reviewer #2: No

5. Review Comments to the Author

Reviewer #1: In this article, women aged 50-69 are invited. But in US GUIDELINES, they recommend 45-74 years women to screen. I think it would be better to screen earlier as breast cancer is quite common before 50 years.

You should give training to women how to carry out breast self examination(BSE),it can help them to be alert to any abnormalities in their breast. Thereby you can assess the high risk group without doing mammography for all.

Because mammography screening may cause harms, the most widely discussed being the over diagnosis. False positive results affecting one in every five women.

Reviewer #2: Review report

Title: Quality assured implementation of the Slovenian breast cancer screening programme

ID: PONE-D-21-04831

Comments: This article comprehends the details of Slovenian breast cancer screening programme. The authors outline various phases of the programme ranging from organization level to implementation stage. As a reviewer, I believe that the effort is notable mainly because it summarizes particulars of a successful nationwide programme targeting a major health concern in women population. The publication of such an article at reputed forum of PLOS will certainly attract wider range of relevant audience. The structure of the programme at patient level is simple, designed, flexible and thus remains workable in different socio-economic backgrounds. For replication purposes, in some other parts of the world, the details of this kind of programme deserve to be publishable (in reviewer’s opinion).

Here are some specific observations of the reviewer.

• Unnecessary lengthy sentences, for example lines 61-67, 121-125 are sample of the issue.

• Sentence structure needed to be more cohesive. As this article details the phases of a very delicate programme; the massage must be very clear and well posed. On the contrary, authors after all the hard work of assembling details of the programme seems little tired at writing stage. Some of the examples are, lines 48-55, 152-155, 208-214 etc. In general authors are trying to comprehend huge amount of information in single sentence. This is observable as a pattern in the writing and seriously compromises the significance of the work.

• The reviewer encourages to use mathematical expressions when necessary. Especially, in data analysis section where vital statistics such as coverage by invitation, programme extension etc. are discussed for the first time.

• It seems better to provide the list of EU performance guidelines to document the performance indicators.

I hope the observations will be helpful in enhancing the quality of the good work conducted by the authors.

6. PLOS authors have the option to publish the peer review history of their article (what does this mean?). If published, this will include your full peer review and any attached files.

Reviewer #1: No

Reviewer #2: No

---

## [Author Response · Author response to Decision Letter 0]

10 Aug 2021

We would like to thank the Reviewers for their thoughtful comments.

We have addressed all of the following comments point by point and have revised the original manuscript using the Track Changes function (see file 'Revised Manuscript with Track Changes').

As requested, we have revised and uploaded the manuscript taking into account the journal requirements and the comments of the reviewers. Please find below copies of the comments received and our respective responses to each issue. 

Sincerely,

Mateja Krajc, corresponding author

mkrajc@onko-i.si

Response: The revised manuscript adheres to the journal’s style requirements.

Once you have amended this/these statement(s) in the Methods section of the manuscript, please add the same text to the "Ethics Statement" field of the submission form (via "Edit Submission").

For additional information about PLOS ONE ethical requirements for human subjects research, please refer to http://journals.plos.org/plosone/s/submission-guidelines#loc-human-subjects-research<http://journals.plos.org/plosone/s/submission-guidelines%23loc-human-subjects-research>.

Response: Screening participants provide informed consent for use of their data for research, and quality assurance of the programme, which includes reporting and publication of aggregate results. Only aggregate, anonymized data was accessed for the preparation of the manuscript. This information has been added to the manuscript (line 98-99: see file 'Revised Manuscript with Track Changes').

Upon re-submitting your revised manuscript, please upload your study's minimal underlying data set as either Supporting Information files or to a stable, public repository and include the relevant URLs, DOIs, or accession numbers within your revised cover letter. For a list of acceptable repositories, please see http://journals.plos.org/plosone/s/data-availability#loc-recommended-repositories. Any potentially identifying patient information must be fully anonymized.

Response: Table 1 contains the minimal data set upon which the present study is based. The data source (Breast Cancer Screening Registry DORA) is stated under the table (line 335).

Reviewers' comments:

Reviewer's Responses to Questions

Comments to the Author

1.-3.

No Response requested.

4. Is the manuscript presented in an intelligible fashion and written in standard English?

Reviewer #1: Yes

Reviewer #2: No

Response: In the finalization of the manuscript, two native English speakers have performed a thorough language revision. These changes are too numerous to mention by line.

5. Review Comments to the Author

Reviewer #1: In this article, women aged 50-69 are invited. But in US GUIDELINES, they recommend 45-74 years women to screen. I think it would be better to screen earlier as breast cancer is quite common before 50 years.

Response: The current age range of the Slovenian Breast Cancer Screening Programme was adopted during the planning phase based on the European Council recommendation on cancer screening published in 2003 (reference No.15 in the manuscript). At the time, this EU policy document was unanimously approved by the health ministers of the EU member states. 

We anticipate that the Slovenian programme will expand the screening age range if an expansion is recommended in the upcoming review of the 2003 EU council recommendation in the framework of the new EU action plan on cancer (https://ec.europa.eu/health/sites/default/files/non_communicable_diseases/docs/eu_cancer-plan_en.pdf).

You should give training to women how to carry out breast self examination (BSE), it can help them to be alert to any abnormalities in their breast. Thereby you can assess the high risk group without doing mammography for all.

Response: In Slovenia BSE is recommended for all women. Nurses in primary care educate women about breast awareness as a means to avoid unnecessary delay in case finding and clinical diagnosis, but not as a substitute for population-based, quality-assured breast cancer screening. The Slovenian screening programme also recommends BSE between two screening examinations. This information is published on the screening website and written in the screening booklet. Letters informing women of their normal screening results also include a recommendation to perform BSE in the interval between two screening mammographies. (lines 184-187).

Because mammography screening may cause harms, the most widely discussed being the over diagnosis. False positive results affecting one in every five women.

Response: We agree that mammography screening may cause harms. The DORA program was set up to minimise harms and maximize benefits through a commitment to quality assurance. In communication with women the programme emphasizes the need to achieve an appropriate balance between the potential harms and benefits of breast cancer screening. This statement has been added to the manuscript (lines 151-153). 

Reviewer #2: Review report

Title: Quality assured implementation of the Slovenian breast cancer screening programme

ID: PONE-D-21-04831

Comments: This article comprehends the details of Slovenian breast cancer screening programme. The authors outline various phases of the programme ranging from organization level to implementation stage. As a reviewer, I believe that the effort is notable mainly because it summarizes particulars of a successful nationwide programme targeting a major health concern in women population. The publication of such an article at reputed forum of PLOS will certainly attract wider range of relevant audience. The structure of the programme at patient level is simple, designed, flexible and thus remains workable in different socio-economic backgrounds. For replication purposes, in some other parts of the world, the details of this kind of programme deserve to be publishable (in reviewer’s opinion).

Here are some specific observations of the reviewer.

• Unnecessary lengthy sentences, for example lines 61-67, 121-125 are sample of the issue.

Response: In the language revision (see our response to Comment 4. above) we have shortened unduly long sentences in the suggested lines.

• Sentence structure needed to be more cohesive. As this article details the phases of a very delicate programme; the massage must be very clear and well posed. On the contrary, authors after all the hard work of assembling details of the programme seems little tired at writing stage. Some of the examples are, lines 48-55, 152-155, 208-214 etc. In general authors are trying to comprehend huge amount of information in single sentence. This is observable as a pattern in the writing and seriously compromises the significance of the work.

Response: Sentence structure has been corrected as suggested.

• The reviewer encourages to use mathematical expressions when necessary. Especially, in data analysis section where vital statistics such as coverage by invitation, programme extension etc. are discussed for the first time.

Response: We have added the formulas for programme extension, and for coverage and participation rates, to the respective column headings in Table 1.

• It seems better to provide the list of EU performance guidelines to document the performance indicators.

I hope the observations will be helpful in enhancing the quality of the good work conducted by the authors.

Response: The full list of EU guidelines performance indicators is provided in reference [6]. Those performance indicators that are regularly monitored by the programme have been added to the text under the heading “Performance indicators” in the “Materials and Methods” section (lines 256-265). In the “Results” section we also explain in greater detail which recommended levels of coverage have been consistently exceeded in recent years (lines 340-344).

---

## [Decision Letter · Decision Letter 1]

27 Sep 2021

Quality assured implementation of the Slovenian breast cancer screening programme

PONE-D-21-04831R1

Dear Dr. Krajc,

We’re pleased to inform you that your manuscript has been judged scientifically suitable for publication and will be formally accepted for publication once it meets all outstanding technical requirements.

Kind regards,

Rashidul Alam Mahumud, MPH, MSc, PhD

Academic Editor

PLOS ONE

Additional Editor Comments (optional):

Reviewers' comments:

Reviewer's Responses to Questions

**Comments to the Author**

1. If the authors have adequately addressed your comments raised in a previous round of review and you feel that this manuscript is now acceptable for publication, you may indicate that here to bypass the “Comments to the Author” section, enter your conflict of interest statement in the “Confidential to Editor” section, and submit your "Accept" recommendation.

Reviewer #2: All comments have been addressed

2. Is the manuscript technically sound, and do the data support the conclusions?

Reviewer #2: Yes

3. Has the statistical analysis been performed appropriately and rigorously? 

Reviewer #2: Yes

4. Have the authors made all data underlying the findings in their manuscript fully available?

Reviewer #2: Yes

5. Is the manuscript presented in an intelligible fashion and written in standard English?

Reviewer #2: Yes

6. Review Comments to the Author

Reviewer #2: Subject: Review Report for the Article entitled “Quality assured implementation of the Slovenian breast cancer screening programme”. Manuscript I. D. PONE-D-21-04831R1

Dear Editor-in-Chief

PLOS ONE,

The quality of the manuscript has significantly improved after revision. I as a reviewer think that the article is consistent with the standards of the Journal. The publication of the details of such public health initiatives, in my opinion, always remain useful on multiple fronts and the usefulness become more noticeable when it is published on reputed forum like PLOS. I recommend this article for publication, and it was a pleasure to read it first hand.

Thanks and Regards

Abdu R Rahman

7. PLOS authors have the option to publish the peer review history of their article (what does this mean?). If published, this will include your full peer review and any attached files.

Reviewer #2: No

---

## [Editor Report · Acceptance letter]

1 Oct 2021

PONE-D-21-04831R1 

Quality assured implementation of the Slovenian breast cancer screening programme 

Dear Dr. Krajc:

I'm pleased to inform you that your manuscript has been deemed suitable for publication in PLOS ONE. Congratulations! Your manuscript is now with our production department. 

Kind regards, 

on behalf of

Dr. Rashidul Alam Mahumud 

Academic Editor

PLOS ONE